# A Voltage-Modulated Nanostrip Spin-Wave Filter and Spin Logic Device Thereof

**DOI:** 10.3390/nano12213838

**Published:** 2022-10-30

**Authors:** Huihui Li, Bowen Dong, Qi Hu, Yunsen Zhang, Guilei Wang, Hao Meng, Chao Zhao

**Affiliations:** 1Beijing Superstring Academy of Memory Technology, Beijing 100176, China; 2Changxin Memory Technologies, Inc., Hefei 230601, China

**Keywords:** spin wave transistor, spin wave filter, voltage modulated

## Abstract

A nanostrip magnonic-crystal waveguide with spatially periodic width modulation can serve as a gigahertz-range spin-wave filter. Compared with the regular constant-width nanostrip, the periodic width modulation creates forbidden bands (band gaps) at the Brillouin zone boundaries due to the spin-wave reflection by the periodic potential owing to the long-range dipolar interactions. Previous works have shown that there is a critical challenge in tuning the band structures of the magnonic-crystal waveguide once it is fabricated. In this work, using micromagnetic simulations, we show that voltage-controlled magnetic anisotropy can effectively tune the band structures of a ferromagnetic–dielectric heterostructural magnonic-crystal waveguide. A uniformly applied voltage of 0.1 V/nm can lead to a significant frequency shift of ~9 GHz. A spin-wave transistor prototype employing such a kind of spin-wave filter is proposed to realize various logical operations. Our results could be significant for future magnonic computing applications.

## 1. Introduction

Magnonic crystals comprising magnetic microstructures distributed periodically, similar to photonic crystal [1], are core components of complex spin logic devices [2,3,4,5,6,7]. Typically, the features of spin waves, such as dispersion relations, are characterized by two different magnetic interactions [6,8]: short-range exchange interactions, which are strong, and long-range dipole–dipole interactions, which are relatively weak. In a uniform nanostrip, the influences of dipole–dipole interactions can be simplified as shape anisotropies [9]. Thus, excited spin waves with short wavelengths are mainly governed by exchange interactions. In the presence of periodic modulation, long-range dipolar interaction gives rise to a periodic potential. As a result, spin waves with specific frequencies get reflected and forbidden bands (band gaps) at the Brillouin zone boundaries are formed [10].

In the past few years, various magnonic crystals have been reported. For example, using micromagnetic simulations, bicomponent magnonic crystals composed of two different materials were fabricated [11,12] and investigated [13,14]. A typical implementation of magnonic crystals is the manufacture of periodic modifications based on regular nanostrips [9,15,16,17,18]. However, the band structures of magnonic crystals cannot be changed once the periodic modification has been fabricated. A uniform magnetic-field bias applied to magnonic crystals can be used to overcome this limit to tune the band structures [19]. The use of a magnetic field, however, is not desirable for device applications. In this work, the band-structure tuning of magnonic crystals by magnetic anisotropy control using electric fields [10,20,21,22] was studied.

Voltage-controlled magnetic anisotropy (VCMA) has been widely studied theoretically and experimentally [20,21]. Interfacial perpendicular magnetic anisotropy can be tuned by applying a voltage without a charge current [22]. The VCMA effect has been used to assist the switching of magnetic tunnel junctions [20] and to control the domain wall traps in ferromagnetic nanowires [21]. Voltage-controlled magnetic anisotropy has also been used to construct reconfigurable magnonic crystals [10], spin-wave nanochannels [23], and logic devices [22]. In this paper, we show that band structures can be tuned using a constant voltage applied to the width-modulated nanostrip. Compared to pure voltage-controlled reconfigurable magnonic crystals [10], the voltage used in this work is uniform and the device is easy to fabricate. Moreover, a relatively small voltage (electric field 0.1 V/nm) can cause a significant shift in frequency (~9 GHz).

## 2. Model and Methods

The width-modulated nanostrip magnonic-crystal waveguide with an ultra-thin cobalt (Co) magnetic layer is illustrated schematically in Figure 1. The entire magnonic-crystal structure is composed of two parts. The left part is a regular nanostrip with a width W_1_ = 30 nm, and the remainder is a periodic-width-modulated nanostrip. The in-plane periodic width modulation is characterized by two typical widths, W1 (30 nm) and W2 (18 nm). The width period is P = P1 (18 nm) + P2 (21 nm) = 39 nm, where P1/P2 represents the segment length of the W1/W2-wide nanostrip. Note that the values of P1 and P2 are assumed here to facilitate gigahertz band-gap simulation. In general, the changing P and P1/P ratios will modify the band gaps [9]. A typical period of tens of nanometers will result in band gaps of a few gigahertz for the dipole-exchange spin waves.

We consider the exchange interactions, uniaxial anisotropy, Zeeman energy, and dipole interactions within the system. Therefore, the total free energy of the studied system can be expressed as:(1)$$E=\int^{\;}\left[A{\left(\nabla m\right)}^{2}-Km_{x}^{2}-\mu_{0}m\cdot h+E_{d}\right]dV$$
where *A* is the exchange constant, **m** is the unit vector of the magnetization, *K* is the uniaxial magnetic anisotropy coefficient, *E_d_* is the demagnetization energy, and **h** is the external excitation field. The effect of the voltage-controlled magnetic anisotropy will be taken into account in the uniaxial anisotropy. 

The magnetization dynamics is described by the Landau–Lifshitz–Gilbert (LLG) equation:$$\frac{\partial m}{\partial t}=-\gamma m\times H_{\mathrm{eff}}+\alpha \;m\times \frac{\partial m}{\partial t}$$
where γ = 2.11 × 10^5^ m/(A·s) is the gyromagnetic ratio, $H_{\mathrm{eff}}=-\frac{1}{\mu_{0}M_{s}}\frac{\delta E}{\delta m}$ is the total effective magnetic field, and α is the Gilbert damping constant. The dispersion relations of the spin-wave excitations are calculated numerically using this equation. We fix α = 0.01 [10] in this work for all the simulations. 

The ground state of an ultrathin Co film is dependent upon the film’s thickness [10]. If the film becomes thinner than the critical thickness, which is typically 0.5–1.0 nm, the ground-state magnetization parallels the out-of-plane axis (z-axis). On the other hand, the magnetization tends to be aligned in-plane if the thickness is larger than the critical thickness. Since VCMA is a pure interface effect, in this study we decided that the thickness of the film would be 1.5 nm with in-plane magnetic anisotropy. 

Micromagnetic simulations using the public Object-Oriented Micromagnetic Framework (OOMMF) [24] were performed. The finite-difference methods were used to compute the total free energy in Equation (1). For this simulation, the following typical parameters of Co were chosen [10]: the exchange constant *A* = 1.5 × 10^−11^ J/m, the saturation magnetization *M_s_* = 5.8 × 10^5^ A/m, and the easy-axis magnetic anisotropy *K* = 0 J/m^3^ for an electric field *E* = 0 V/nm. The cell geometry was 1.5 nm × 1.5 nm × 1.5 nm, below the studied material’s exchange length. 

We computed the band structures of the magnonic crystals by performing a 2D discrete Fourier transform for the temporal and spatial magnetization data, i.e., *m_y_* along the x-axis at *y* = 15 nm, collected every 1 ps for a duration of 8 ns, applying an excitation field to the system. The external field reads [25]:$$h=h_{0}\mathrm{sinc}\left(\omega_{c}\left(t-t_{0}\right)\right)\mathrm{sinc}\left(k_{c}\left(x-x_{c}\right)\right)\overset{w/dy}{\underset{i=1}{\sum }}\sin\frac{i\pi y}{w}e_{y}$$
where *h*_0_ = 1000 A/m, sinc(x) = sin(πx)/(πx), *t*_0_ = 50 ps, *ω_c_* = 120 GHz, *k_c_* = 0.1 nm^−1^, *x_c_* = 1500 nm, *w* is the width of the wire, and *dy* = 1.5 nm. Both symmetric and antisymmetric modes of spin waves [25] were excited by the above-mentioned magnetic field signal.

## 3. Results and Discussion

Figure 2 shows the calculated dispersion curves for the spin-wave excitation along the x-axis at *y* = 15 nm. In Figure 2a, spin waves below 4.6 GHz are not allowed to propagate, forming an intrinsic forbidden band. This is because the dispersion relation of spin waves for the nanotrack can be written as:$$\omega =\left(2\gamma /\mu_{0}M_{s}\right)\sqrt{\left(K+{\mathrm{Ak}}^{2}\right)\left(K+K_{⊥}+{\mathrm{Ak}}^{2}\right)}$$
where *K* and *K*_⊥_ are two effective anisotropies. Therefore, the minimum allowed frequency $\omega_{c}=\left(2\gamma /\mu_{0}M_{s}\right)\sqrt{K\left(K+K_{⊥}\right)}$. Owing to the periodic modifications, the lower band gaps ∆*_l_* = 12.6–19.2 GHz emerge at the Brillouin zone (BZ) boundaries, denoted by the blue dashed vertical lines. The BZ boundaries are located at the positions *k_x_* = (2n + 1)π/P, where n = 0, ±1, ±2, … and P is the width period of the magnonic crystals. The higher band gaps ∆*_h_* = 37.6–44.2 GHz are found at *k_x_* = 2nπ/P, which are represented by the red dotted vertical lines. Clearly, both the lower and higher band gaps are associated with the one-dimensional translation symmetry of the shape periodicity in the x-direction.

In general, the magnonic-crystal band structures were influenced by W_1_ (W_2_), P_1_ (P_2_), and P. In this work, we fixed all of them and studied the influence of applied voltage. We assumed that the strength of the surface anisotropy energy varied linearly with the electric field *E*, that is, ∆*K*_s_ = β*_s_E*, where β*_s_* is the magnetoelectric coefficient. Various magnetoelectric coefficients have been reported [10,20,23], and in our simulation we used β_s_ = 100 fJ/Vm. Moreover, we assumed that *K*_s_ = 0 when the applied voltage was zero. Accordingly, an electric field of 0.1 V/nm corresponds to a surficial anisotropy energy of *K*_s_ = 0.1 mJ/m^2^ and thus an effective ∆*K* = $K_{s}/t_{c}$ = 6.67 × 10^4^ J/m^3^, where $t_{c}$ is the thickness of the sample. Figure 2b shows the corresponding dispersion curves when an electric field of 0.1 V/nm was applied. It is clearly shown that the intrinsic forbidden gap was shifted to 13.6 GHz due to the applied voltage (*E* = 0.1 V/nm). The lower and higher band gaps were changed to ∆*_l_* = 19.9–26.2 GHz and ∆*_h_* = 44.6–51.2 GHz. Therefore, a relatively small voltage can tune the band structures effectively. It is worth mentioning that magnetoelectric coefficients of about 100 fJ/Vm in a CoFeB/MgO [26] system and over 300 fJ/Vm in an iridium-doped Fe/MgO [27] system have been reported. Both material combinations are also promising for VCMA-modulated magnonic crystal applications.

Dispersion curves for applied electric fields of 0.05 V/nm and 0.15 V/nm were also calculated. Figure 3 summarizes the typical features of all the simulated dispersion curves. As can be seen in Figure 3a, the three typical frequencies changed nearly linearly with the applied electric fields. Frequency F_1_ measured the intrinsic forbidden band, which corresponded to *k_x_* = 0. Frequencies F_2_ and F_3_ determined the lower band gap, i.e., ∆*_l_ =* F_2_ − F_3_. All of the three typical frequencies increased as the electric field increased. Meanwhile, their incremental amplitudes were different, as shown in Figure 3b. For example, the ∆F for F_1_ was the largest, although the value of frequency F_1_ was the lowest. The tunabilities of the applied electric fields for F_2_ and F_3_ were almost the same, indicating a stable forbidden band-gap width.

Here, we give an example to explain the electric-field-modulated spin-wave filter effect in the width-modulated nanostrip waveguide. As shown in Figure 2, the band structures of the spin-wave excitations in the width-modulated magnonic crystals without (left) and with (right) an applied electric field of 0.1 V/nm were compared. Two typical frequencies of 16 GHz and 22 GHz were selected for analysis. Without any electric voltage applied, the spin wave of 16 GHz was in the forbidden band gap and the spin wave of 22 GHz was in the allowed band. That means, intrinsically, that the 16 GHz spin-wave signal could be filtered by this width-modulated nanostrip waveguide. However, after applying an electric field of 0.1 V/nm, the spin wave of 16 GHz was in the allowed band and the spin wave of 22 GHz was in the forbidden band. The spin-wave filtering effect was reversed by the electric field. Therefore, the width-modulated nanostrip magnonic-crystal waveguide can serve as a spin-wave filter [15], and its filtering effect can be tuned by electric fields.

To verify the electric-field-tunable spin-wave filtering effect, we excited spin waves using a *sine* field **h** = *h*_0_sin(2πft)**e**_y_ locally in the region *x* ≤ 3 nm, with *h_0_* = 10,000 A/m. Figure 4a,b show the snapshots of the intrinsic spin-wave propagations without an electric field in a width-modulated nanostrip magnonic-crystal waveguide with excitation frequencies of *f* = 16 GHz and *f* = 22 GHz, respectively. The two frequencies selected are typical for the forbidden band and the allowed band, respectively. The red and blue colors represent the z-component of magnetization, *m_z_*. At *t* = 0.2 ns, spin waves formed in the left regular region of the nanostrip. For the *f* = 22 GHz case, the spin waves propagated forward, as can be seen at *t* = 0.3 ns, 0.4 ns, and 0.5 ns. However, as a comparison, spin waves failed to move into the width-modulation region when the excitation frequency *f* = 16 GHz, as shown in Figure 2, indicating the existence of the forbidden gaps. 

Figure 5 shows the spin-wave propagation snapshots with an electric field of 0.1 V/nm and excitation frequencies of *f* = 16 GHz and *f* = 22 GHz, respectively. In this situation, the frequency *f* = 22 GHz was located in the forbidden band, while the spin waves with frequency *f* = 16 GHz were allowed to pass through the width-modulated region, as shown in Figure 2.

In Figure 6, an application prototype of a spin-wave transistor using a width-modulated magnonic-crystal spin-wave filter is presented. The regular uniform-width magnetic nanostrip is used as a source terminal (spin-wave injection) and a drain terminal (spin-wave output) for the proposed spin-wave transistor to propagate spin waves. The VCMA system is composed of a metal gate layer, a dielectric layer, and a magnonic-crystal layer that plays a role as a transistor channel to filter the spin waves. The electric voltage applied to the gate can control the spin-wave propagation channel’s opening and closing, thus realizing the binary code output of 1 and 0, respectively. The spin-wave transistor proposed here can be used as a fundamental building block for future magnonic computing systems to realize various spin logic operations. 

## 4. Conclusions

The voltage-controlled spin-wave filtering behavior in width-modulated nanostrip magnonic-crystal waveguides was studied using micromagnetic simulations. The band structures of the magnonic-crystal waveguide cannot be varied once the periodic modification is fabricated. We have shown that a uniform voltage is sufficient to tune the band structures and that a relatively small voltage with an electric field of 0.1 V/nm can lead to a significant frequency shift (~9 GHz). A spin-wave transistor prototype was finally proposed based on the VCMA-tunable magnonic-crystal spin-wave filter.

## Figures and Tables

**Figure 1 nanomaterials-12-03838-f001:**
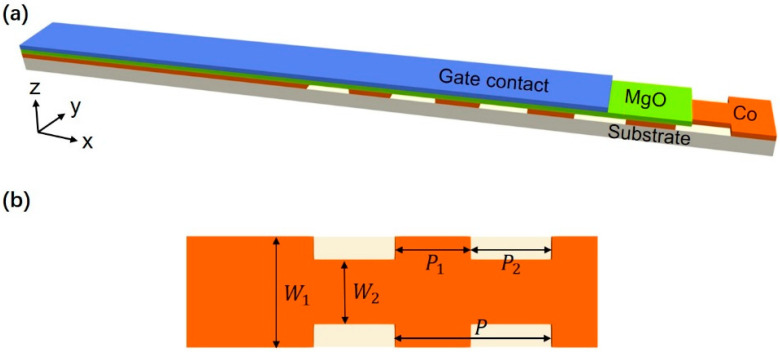
(**a**) Schematic illustration of the width-modulated nanostrip magnonic-crystal waveguides. The film stack is a metal/MgO/Co (1.5 nm)/insulate substrate. The top layer is the gate metal. A voltage was applied on the gate contact to tune the anisotropy. The Co layer is fabricated with periodic width modulations. (**b**) Four parameters characterize the geometry of the Co layer: the width of the nanostrip, W_1_, and a second width at the narrow place, W_2_; the lengths of the segments with widths W1 and W2 are P_1_ and P_2_, respectively. In the simulation, we used W_1_ = 30 nm, W_2_ = 18 nm, P_1_ = 18 nm, and P_2_ = 21 nm.

**Figure 2 nanomaterials-12-03838-f002:**
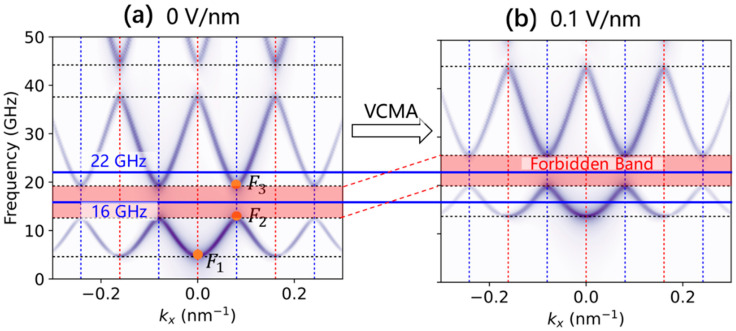
(**a**) Dispersion curves of the spin-wave excitations in the width-modulated nanostrip magnonic crystals. The dispersion curves were calculated using a 2D discrete Fourier transform for the temporal and spatial data at the wire center (*y* = 15 nm), which were collected every 1 ps after applying an external magnetic field. Lower and higher band gaps of ∆*_l_* = 12.6–19.2 GHz and ∆*_h_* = 37.6–44.2 GHz were observed due to the periodic width modulations. (**b**) The corresponding dispersion curves when applying an electric field of *E* = 0.1 V/nm. Due to the VCMA-induced anisotropy, the band gaps changed to ∆*_l_* = 19.9–26.2 GHz and ∆*_h_* = 44.6–51.2 GHz. The forbidden band gap is filled in pink. Selected typical frequencies of 16 GHz and 22 GHz are labelled using blue lines to guide the eyes.

**Figure 3 nanomaterials-12-03838-f003:**
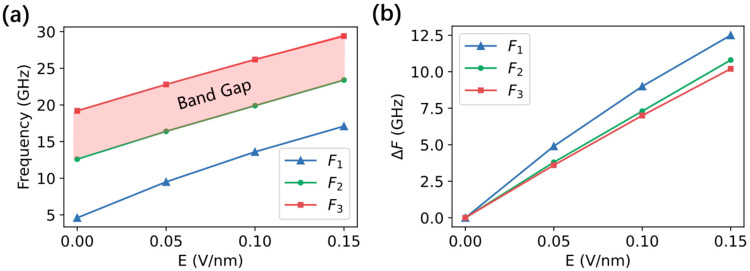
(**a**) The dependence of typical frequencies on applied electric fields. F1 was the lowest allowed frequency which occurred at *k_x_* = 0. F2 and F3 determined the lower band gaps at the first BZ boundary, where *k_x_* = π/P. (**b**) The corresponding incremental frequencies for F_1_, F_2_, and F_3_.

**Figure 4 nanomaterials-12-03838-f004:**
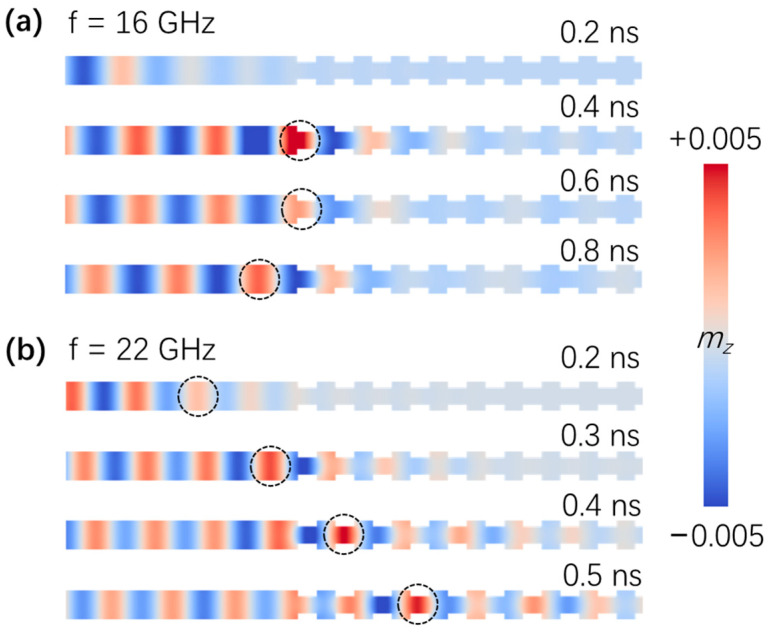
(**a**) Spin-wave propagation along the width-modulated nanostrip magnonic-crystal waveguide without electric field. The spin waves were excited using a sine field **h** = *h*_0_sin(2πft)**e**_y_ locally in the region *x* ≤ 3 nm, with *h*_0_ = 10,000 A/m and frequency *f* = 16 GHz. The dashed circles are shown to guide the eyes. The spin waves could not pass through the narrow part with width modulations. (**b**) As a comparison, the frequency of the excitation field *f* = 22 GHz was plotted. Clearly, the spin waves propagated even in the presence of width modulations, as the red area moved towards the right with time. A linear color map is used in plotting the m_z_.

**Figure 5 nanomaterials-12-03838-f005:**
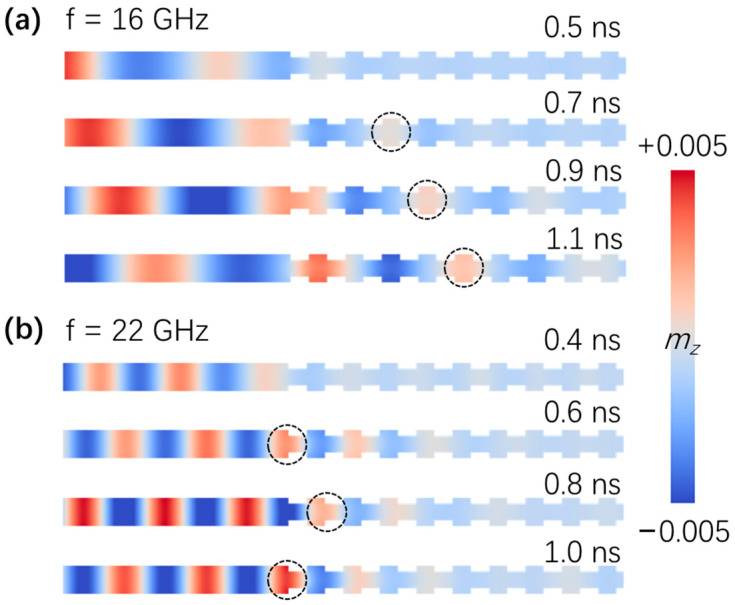
(**a**) Spin-wave propagation along the width-modulated nanostrip with an electric field of 0.1 V/nm. The frequency of the excitation field was *f* = 16 GHz, which was not in the band gaps. As expected, the spin waves moved forwards along the waveguide. (**b**) The spin waves failed to pass through the width-modulated region when the excitation frequency was *f* = 22 GHz with the applied electric field.

**Figure 6 nanomaterials-12-03838-f006:**
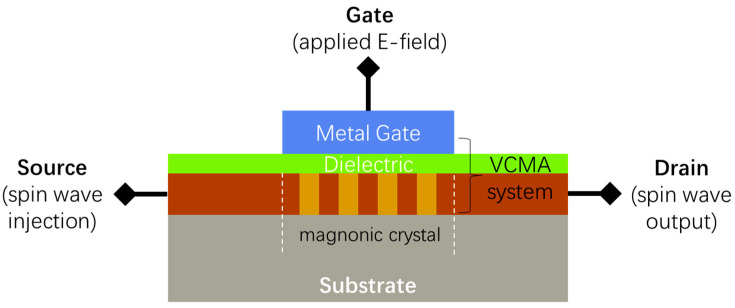
Schematic illustration of a spin-wave transistor proposal based on a VCMA-controlled spin-wave filter using width-modulated magnonic crystals. The VCMA system composed of a metal gate layer, a dielectric layer, and a magnonic-crystal layer was adapted to allow (binary code 1) or forbid (binary code 0) the propagation of spin-wave signals.

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
