# Peer review of "A Voltage-Modulated Nanostrip Spin-Wave Filter and Spin Logic Device Thereof"

_nanomaterials, 2022, doi:10.3390/nano12213838_

Round 1
Reviewer 1 Report
The paper contains an interesting micromagnetic study of spin waves propagation in a magnonic crystal realized by a voltage-modulated nanostrip. It is shown that the voltage-controlled magnetic anisotropy can effectively tune the band structure of the designed magnonic-crystal waveguide and that an applied voltage of 0.1 V/nm can lead to a significant frequency shift of the band of spin waves, so that such a kind of filter is proposed to realize various magnonic computing applications. The idea of the paper is interesting and the methodology appropriate, so I think that the paper deserves publicarion in Nanomaerials. However, the following points shoul be fixed before pubblication.
1) In order to make the paper more interesting for applications, the authors should clarify in more details which is the material combination that would permit, in reality, to obtain the results they show applying a small voltage (for instance at row 128 in paragraph 3).
2) In the caption of Fig. 1 the parameters P1 and P2 are called "periods", but this is not correct. It would be better to call them "widths".
3) In Fig. 2 the right and left panel should have the same scale on the vertical axis, so that the frequencies can be directly compared by the reader. So, please, plot both panels between 0 and 50 GHz (or between 0 and 60 GHz).
4) Fig. 4 is redundant and can be sufficient to indicate the forbidden bands in Fig. 2.
5) In Figs. 5 and 6 it would be interesting to know if the color scale on the right is linear or logarithmic, so please clarify in the caption or put numbers.
Reviewer 2 Report
1. This is about the results of micromagnetic simulations of a nanostrip magnonic-crystal waveguide with spatially periodic width modulation, which may serve as a gigahertz-range spin-wave filter as they claim.
2. In the simulation, they used W1 = 30 nm, W2 = 18 nm, P1 = 18 nm, P2 = 21 nm. Why did they choose these specific parameters? Appropriate explanation should be given.
3. Using the micromagnetic simulations, they calculated the dispersion curves of the spin-wave excitations in the width-modulated nanostrip magnonic-crystal under various strength of the electric fields.
4. From the results of the simulations, they suggested spin-wave filtering by the applied electric filed (0.1 V/nm) in the two typical frequencies of 16 GHz and 22 GHz.
5. The main advantage of their proposal “the voltage-controlled width-modulated nanostrip magnonic-crystal waveguides” seems to be that a significant frequency shift (∼ 9 GHz) can be achieved by simply applying the relatively small electric field (0.1 V/nm).
Reviewer 3 Report
Dear Authors,
in your interesting manuscript, the following points should be added/changed to further improve it:
- section 2: Please give the reader an idea why these dimensions were chosen (due to previous tests?) and what would change with changing them.
- line 99: What is meant with sinc = sin(πx)/(πx) - is it "sinc(x) = ..."?
- line 146: "GHz" is included in F2 and F3 and thus not necessary here.
- line 155: the black line is blue
- Generally, please describe more in detail how you performed the OOMMF simulations. The general magnetic part is clear, but how do you insert the voltage?
